# Molecular detection and antimicrobial resistance profiles of Extended-Spectrum Beta-Lactamase (ESBL) producing *Escherichia coli* in broiler chicken farms in Malaysia

Mulu Lemlem[1,2]*, Erkihun Aklilu[1]*, Maizan Mohammed[1], Fadhilah Kamaruzzaman[1], Zunita Zakaria[3], Azian Harun[4], Susmita Seenu Devan[1]

1 Faculty of Veterinary Medicine, Universiti Malaysia Kelantan, Kota Baharu, Malaysia, 2 Department of Medical Microbiology and Immunology, College of Health Science, Mekelle University, Tigray, Ethiopia, 3 Faculty of Veterinary Medicine, Universiti Putra Malaysia, Serdang, Malaysia, 4 School of Medical Sciences, Universiti Sains Malaysia, Kubang Kerian, Malaysia

* mululemlem86@gmail.com (ML); erkihun@umk.edu.my (EA)

## Abstract

Antimicrobial resistance is one of the major public health threats globally. This challenge has been aggravated with the overuse and misuse of antibiotics in food animals and humans. The present study aimed to investigate the prevalence of Extended-Spectrum β-lactamase (ESBL) genes in *Escherichia coli* (*E. coli*) isolated from broiler chickens in Kelantan, Malaysia. A total of 320 cloacal swabs were collected from farms in different districts of Kelantan and were analyzed using routine bacteriology, antimicrobial susceptibility test, and molecular techniques for further identification and characterization of ESBL encoding genes. Based on PCR detection for the *E. coli* species-specific *Pho* gene, 30.3% (97/320) of isolates were confirmed as *E. coli*, and 84.5% (82/97) of the isolates were positive for at least one ESBL gene. Majority of the isolates, 62.9% (61/97) were harboring $bla_{CTX-M}$ followed by 45.4% (44/97) of $bla_{TEM}$ genes, while 16.5% (16/97) of the isolates were positive for both *mcr-1* and ESBL genes. Overall, 93.8% (90/97) of the *E. coli* were resistant to three or more antimicrobials; indicating that the isolates were multi-drug resistance. 90.7% of multiple antibiotic resistance (MAR) index value greater than 0.2, would also suggest the isolates were from high-risk sources of contamination. The MLST result shows that the isolates are widely diverse. Our findings provide insight into the alarmingly high distribution of antimicrobial resistant bacteria, mainly ESBL producing *E. coli* in apparently healthy chickens indicating the role of food animals in the emergence and spread of antimicrobial resistance, and the potential public health threats it may pose.

## Introduction

Antimicrobial resistance (AMR) is one of the most challenging threats globally. Excessive use of antimicrobials in veterinary medicine, food-animal production, and agriculture results in the emergence of antimicrobial resistance [1–3]. Modern food animal production system

**Data Availability Statement:** All relevant data are within the manuscript and its Supporting Information files.

**Funding:** The authors would like to thank the Faculty of Veterinary Medicine, Universiti Malaysia Kelantan for providing research facility to conduct the research. The laboratory investigation of this research was funded by Ministry of Higher Education Malaysia (MOHE) under the fundamental research grant scheme (FRGS), grant no. R/FRGS/A0600/00553A/005/2019/00700 awarded to Dr Erkihun Aklilu. The funders had no role in study design, data collection and analysis, decision to publish, or preparation of the manuscript.

**Competing interests:** The authors have declared that no competing interests exist.

requires large amounts of antimicrobials for disease control, prophylaxis, and growth promotion. Resistance among bacterial species causes increment in morbidity, mortality, and treatment costs worldwide [4]. Multi-drug resistant bacterial infections can result in minimum treatment choices, thus threats of AMR have reached an alarming level [5].

One of the mechanisms of bacterial resistance to antibiotics is producing an enzyme that hydrolyze the beta-lactam ring of antibiotics. Extended-spectrum β-lactamases are highly potent new bacterial enzymes that are resistant to β-lactam antibiotics [6]. ESBLs are resistant to the three generations of cephalosporin, whereas they might be inhibited by β-lactamase inhibitors specifically clavulanic acid [7]. ESBLs were reported for the first time in 1980s and are responsible for causing nosocomial and community acquired infections. ESBLs are plasmid-encoded enzymes commonly found in *Enterobacteriaceae*, mainly in *E. coli*, *Klebsiella* and *Salmonella* [8]. *E. coli* is usually the harmless facultative anaerobic bacteria, which is mostly found in the gastrointestinal tract of humans and animals. However, it is also a potentially pathogenic bacteria that can cause various diseases and is considered as a main cause of mortality and morbidity in poultry farms [9, 10]. Resistant bacteria in food animals may directly or indirectly be transferred to humans through food, water, and manure. Due to the ubiquitous nature of the commensal bacteria, they are reservoirs of resistance determinants [11]. Fecal carriage of extraintestinal pathogenic *E coli* (EXPEC) associated genes in chicken is found to cause EXPEC infections in animal model [12]. In addition, recent evidences showed that a part of human food born EXPEC infections originated from food producing animals mainly poultry meat [13]. Therefore, identifying these resistant *E. coli* isolates from apparently healthy food animals is important to understand the antibiotic resistance characteristics, the emergence and spread of resistance genes, particularly ESBL encoding genes. ESBL is an increasing threat for the public health in developing countries including Malaysia. Chicken meat and its products are the main source of protein in Malaysia [14]. There are several reports on the prevalence of ESBL producing *E. coli* in health sectors in Malaysia [15–17]. In addition, recent evidence has shown that there is a high contamination of chicken meat with ESBL producing *E. coli* in Malaysia [18–20]. Even though there is research done in some parts of the peninsular Malaysia on ESBL producing *E. coli* from chicken farm, but there is no published data on the prevalence of ESBL producing *E. coli* and its encoding genes from broiler chicken farm in the study area. Therefore, this study was aimed to investigate the occurrence of ESBL producing *E. coli*, the resistance genes and antimicrobial resistance patterns of *E. coli* isolated from apparently healthy broiler chickens in Kelantan.

## Methodology

### Sample collection

A total of 320 cloacal swab samples were randomly collected from 5 different broiler farms namely, Machang, bachok, Tumpat, Pasir Mas and Jeli in Kelantan. Each cloacal swab was placed into Amies transport medium and labeled with sample identification number and date of collection. All collected samples were transported to the laboratory using an ice pack at a temperature of 2–8˚C within 6 hours of sample collection.

### Isolation and identification of *E. coli*

Collected cloacal swab samples were enriched in Buffered Peptone Water (Oxoid, Manchester, UK) and incubated at 37˚C for 24 hrs. The enriched bacteria were inoculated on MacConkey agar and lactose-fermenting colonies were taken and cultured on Eosin methylene blue (EMB) agar (Oxoid, Manchester, UK) after incubation at 37˚C for 24 hrs. Green metallic sheen colonies on EMB were selected for further confirmation by using biochemical characteristics,

including triple sugar iron agar (TSI) for glucose fermentation, citrate utilization, urease production, indole fermentation, methyl red test, and motility as mentioned previously [21]. Presumptive *E. coli* isolates were sub cultured on to nutrient Agar and stored in Luria-Bertani (LB) broth (Oxoid, Manchester, UK) containing 50% glycerol at -80˚C for further analysis as described in [22]. *E. coli* ATCC® 25922 was used as a positive control. Isolates were confirmed by PCR, using a set primer specific for *E. coli*.

## Antimicrobial susceptibility test

Antimicrobial susceptibility testing (AST) of *E. coli* isolates was performed using the Kirby-Bauer disk diffusion method on Mueller-Hinton agar (MHA) (Oxoid, Manchester, UK). Bacterial suspension with turbidity equivalent to 0.5 McFarland standard was evenly dispensed on the surface of MHA plates using a sterile cotton swab. Antibiotic discs Aztreonam (ATM30), Cefotaxime (CTX30), Amoxicillin-clavulanic acid (AMC30), Ceftazidime (CAZ30), Ceftriaxone (CRO30), Trimethoprim-sulfamethoxazole (SXT25), Chloramphenicol (C30), Tetracycline (TE30), Tazobactam (TZP110), Ofloxacin (OFX5), Imipenem (IPM10) and meropenem (MEM10) were placed on the surface of MHA agar plates and incubated at 37˚C for 16 to 18 hours. The zone of inhibition was measured to the nearest millimetre and interpreted based on the guidelines of Clinical and Laboratory Standards Institute (CLSI) [23]. *E. coli* ATCC® 25922 were used as a control strain. Bacterial isolates that show resistant to three or more classes of antimicrobial agents were classified as multidrug resistant (MDR) [24, 25]. Multiple antibiotic resistance (MAR) index was analyzed as stated in [26]. Multiple antibiotic resistance (MAR) index was calculated by dividing the number of resistant antibiotics which are resistance to an organism to the total number of antibiotics tested.

$$\text{MAR index} = \frac{\text{Number of antimicrobials to which the isolate showed resistance}}{\textit{Number of total antibiotics exposed to the isolate}}$$

## PCR confirmation and ESBL encoding gene detection in *E. coli*

Genomic DNA was extracted using bacterial DNA extraction kit (Machery-Nagel, Germany) following the manufacturer's recommendation. Extracted DNA was amplified using PCR with species-specific Pho and E. coli primer as published previously [18, 19, 27, 28]. The primer sequences used in this study are summarized in Table 1. The PCR reaction for pho primer was carried out with the following protocol: An initial denaturation step of 95˚C for 4 min followed by 30 cycles of denaturation at 95˚C for 30 s, optimized annealing temperature at 56˚C for 30 s and extension at 72˚C for 60 s with a final extension at 72˚C for 10 min. The PCR protocol for

**Table 1. Primers used for the detection of *E. coli* species and ESBL genes.**

| Target gene | Primer sequence | Amplicon size (bp) | Annealing temperature | Reference |
|---|---|---|---|---|
| **Alkaline Phosphatase (*Pho A*)** | F: 5′– GTG ACA AAA GCC ACA CCA TAA ATG CCT-3′ | 903 | 56 | [18, 27] |
| | R: 3′-TAC ACT GTC ATT ACG TTG CGG ATT TGG CGT-5′ | | | |
| *E. coli* | F: 5'-TGACGTTACCCGCAGAAGAA-3' | 832 | 55 | [19] |
| | R: 3'-CTCCAATCCGGACTACGACG-5' | | | |
| ***bla*CTX-M** | F: 5' -ATG TGC AGY ACC AGT AAR GTK ATG GC-3' | 592 | 60 | [29] |
| | R: 3'- TGG GTR AAR TAR GTS ACC AGA AYS AGC GG -5' | | | |
| ***bla*TEM** | F:5′-GCG GAA CCC CTA TTT G | 964 | 55 | |
| | R: 3'-ACC AAT GCT TAA TCA GTG AG-5' | | | |
| ***mcr-1*** | F: 5'-AGTCCGTTTGTTCTTGTGGC-3' | 320 | 58 | [30] |
| | R: 3'-AGATCCTTGGTCTCGGCTTG-5' | | | |

E coli primer was as follows: Initial denaturation of 95˚C for 3 min; 35 cycles of denaturation at 95˚C for 15 sec, annealing at 55˚C for 90 sec and extension at 72˚C for 15 sec followed by final extension at 72˚C for 10 min. Extracted genomic DNA was further amplified using PCR with specific primers (Table 1) to screen for the presence of ESBL ($bla_{CTX-M}$ and $bla_{TEM}$) and colistin (*mcr-1*) encoding genes in *E. coli*. The PCR Protocol used for both CTX and TEM: an initial denaturation 95˚C for 4 min, followed by 30 cycles of 94˚C denaturation for 30s, 55˚C annealing temperature for 30s and extension 72˚C for 60 sec, with the final extension 72˚C for 10 min. the Agarose gel electrophoresis of the PCR products were conducted, and gel images were analysed using GelDoc$^©$ Gel Documentation System (Bio-Rad, USA).

A correlation heatmap was generated between the antibiotic resistance genes and resistance phenotypes using a Python seaborn library for categorical variables [31]. The phenotype resistance of the antibiotics was determined based antibiotic susceptibility profiles.

## Multi-Locus sequence typing (MLST)

Seven housekeeping genes, *adk*, *fumC*, *gyrB*, *icd*, *mdh*, *purA*, and *recA*, were amplified and sequenced for each selected isolates following previous protocol [32, 33].The primers used are available online in website [34]. The PCR conditions were as follows: initial denaturation at 95˚C for 2 min; 30 cycles of 95˚C for 1 min, 56˚C (*adk*) or 64˚C (*fumC*, and *purA*) or 68˚C (*recA*, *gyrB*, *icd* and *mdh*) for 1 min and 72˚C for 2 min; followed by a final extension step at 72˚C for 5 min. The amplified PCR products were sent to Apical 1st base Sequencing service (Apical Scientific SDN. BHD.) Malaysia, to perform a sequence analysis. The allele sequences and sequence types were determined from the *E. coli* database at the MLST website [35].

## Ethical approval

This study was approved by the Institutional Animal Care and Use Committee of Universiti Malaysia Kelantan (Approval code: UMK/FPV/ACUE/PG/2/2019, Approval Date: February 2019). The animal subjects (chicken from commercial poultry farms) were used only for cloacal swabs collection and no invasive or harmful procedures were used in handling the birds.

## Result

In this study, out of 320 cloacal swab samples, 121 presumptive *E. coli* were isolated based on routine bacteriological and biochemical characteristics. Out of these 121 *E. coli* isolates, from Machang were (n = 35), Bachok (n = 28), Tumpat (n = 20), Pasir Mas (n = 21) and Jeli (n = 17). Out of these *E. coli* isolates, 30% (97/320) were confirmed as *E. coli* by PCR using species-specific primer.

## Antimicrobial susceptibility profile

Antimicrobial susceptibility pattern of the *E. coli* isolates against 12 antibiotics of different classes were subjected and the results are summarized in Table 2. *E. coli* isolates had relatively higher resistance to tetracycline (82.5%) and sulfamethoxazole/trimethoprim (78.4%), whereas the lowest resistance (17.5%) was observed to the imipenem as shown in Table 2. In addition, some of the *E. coli* isolates were resistant to carbapenem antibiotics, meropenem (28.9%) and imipenem (17.5%). Almost all, 93.8% (91/97) of the isolates were resistant to at least three of the tested 12 antibiotic discs belong to different classes. Thus, they could be classified most of the *E. coli* isolates of this study are multi-drug resistant. 90.7% of the isolates were with MAR index values > 0.2; while 9.3% of them had index values less than or equal to 0.2. The

**Table 2. Antimicrobial resistance profiles of *E. coli* isolate from chicken farm in Kelantan, Malaysia, 2021 (n = 97).**

| Antibiotic | Antibiotic class | R | R (%) |
|---|---|---|---|
| Imipenem (IPM10) | Carbapenem | 17 | 17.5 |
| Meropenem (MEM10) | Carbapenem | 28 | 28.9 |
| Tazobactam (TZP110) | Beta-lactamase inhibitor | 36 | 37.1 |
| Ceftazidime (CAZ30) | Cephalosporin | 46 | 47.4 |
| Aztreonam (ATM30) | Beta-lactam | 49 | 50.5 |
| Cefotaxime (CTX30) | Cephalosporin | 51 | 52.6 |
| Ofloxacin (OFX5) | Quinolones | 58 | 59.8 |
| Amoxicillin-clavulanic acid (AMC30) | Beta-lactamase inhibitor | 59 | 60.8 |
| Ceftriaxone (CRO30) | Cephalosporin | 60 | 61.9 |
| Chloramphenicol(C30) | Chloramphenicol | 60 | 61.9 |
| Trimethoprim-sulfamethoxazole (SXT25) | Sulphonamides | 76 | 78.4 |
| Tetracycline (TE30) | Tetracycline | 80 | 82.5 |

proportions of isolates with the MAR index values from 0.3, 0.4, 0.5, 0.6, 0.7, 0.8, and 0.9 were 16.5%,16.5%,7.2%, 15.5%,10.3%, 12.4% and 8.3% respectively.

## ESBL encoding genes in *E. coli* isolates

The *E. coli* isolates were screened for the presence of ESBL encoding genes where $bla_{CTX-M}$ and $bla_{TEM}$ genes were detected in 62.9% (61/97) and 45.4% (44/97) of the samples respectively. In addition, 20.6% (20/97) were positive for colistin resistance gene *mcr-1*. Our study showed that 22.7% (22/97) of the *E. coli* isolates were harboring both $bla_{CTX-M}$ and $bla_{TEM}$ genes. The majority the ESBL resistance genes found were $bla_{CTX-M}$ gene. Six isolates (6.1%) were found positive for $bla_{CTX-M,}$ $bla_{TEM}$ and *mcr-1* genes. Overall, 84.5% (82/97) of those *E. coli* isolates were positive for ESBL genes. The ESBL and colistin resistance genes distribution are summarized in Table 3.

## Multi-Locus sequence typing (MLST)

The MLST sequence typing of the isolates shows that the *E. coli* isolates were widely diverse.

**Table 3. MLST sequence type of *E. coli* isolates and encoding resistance genes.**

| Sample ID | ST | ST complex | ESBL Gene |
|---|---|---|---|
| KT10 | ST93 | ST168CPLX | $bla_{CTX}$ |
| Cs8 STX | ST540 | | *mcr-1* |
| KT15 | ST373 | ST168copmlex | *mcr-1* |
| KT18 | ST69 | ST69 complex | $bla_{TEM}$ |
| KT9 | ST154 | | *mcr-1* |
| KT3R | ST93 | ST168comlex | *mcr-1* |
| KT22R | ST226 | ST226 complex | $bla_{TEM}$ |
| Cs8 | ST117 | | $bla_{CTX}$, $bla_{TEM}$ |
| KT33 | ST345 | | $bla_{CTX}$, $bla_{TEM}$, *mcr-1* |
| CS34 | ST196 | | $bla_{TEM}$, *mcr-1* |
| CS32STK | ST1001 | | $bla_{CTX}$, $bla_{TEM}$, *mcr-1* |
| KT23-26/2 | ST155 | ST155cplx | $bla_{CTX}$, $bla_{TEM}$, *mcr-1* |
| KT32R | ST345 | | $bla_{CTX}$, $bla_{TEM}$, *mcr-1* |
| JL24 | ST1638 | ST10cplx | $bla_{CTX}$, $bla_{TEM}$, *mcr-1* |

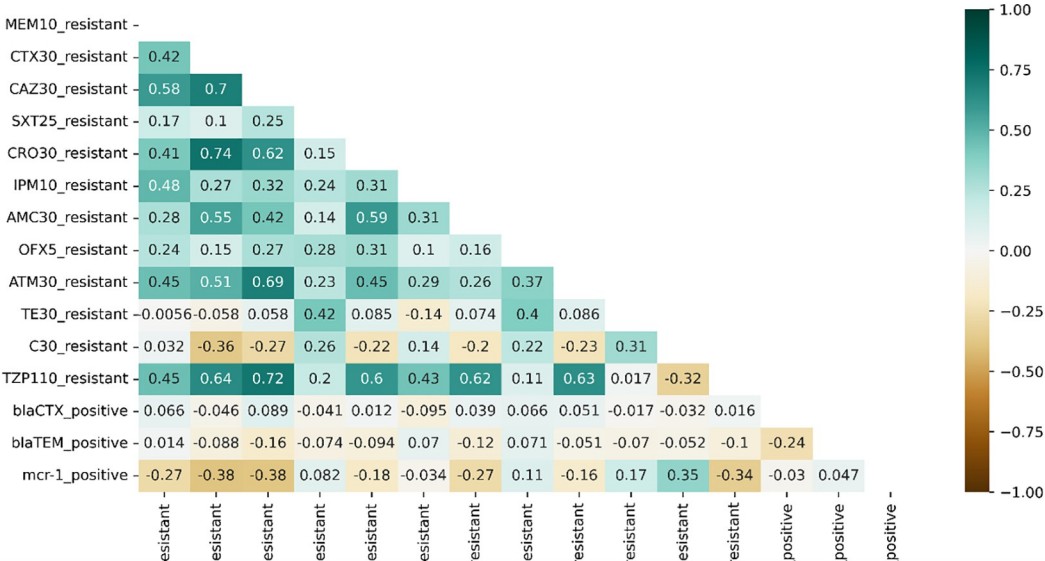

**Fig 1. Correlation heatmap for ESBL genes with antibiotic resistant phenotypes.**

The Identified sequence types are summarized in Table 3.

The correlation coefficient of the phenotype resistant to ESBL Encoding ($bla_{CTX-M}$ and $bla_{TEM}$) genes of the isolated *E. coli* shows near to zero value (Fig 1). However, the a few resistance antibiotics, for instance, CAZ30 and CTX30; CRO30 and CTX30; ATM30 and CAZ30; TZP110 and CAZ30 have shown a strong correlation pattern.

## Discussion

Even though *E. coli* is a commensal bacterium, it is the main opportunistic pathogen in poultry. ESBL producing *E. coli* pose major threat to poultry production with a potential risk of transfer of these resistant pathogens to humans directly or indirectly. In the present study, ESBL encoding genes of *E. coli* were detected from cloacal swab samples collected from broiler chickens in Kelantan, Malaysia. The 30% prevalence of *E. coli* among the samples collected from the broiler chicken farm in the current study is lower than the previous study from Malaysia which reported a prevalence of 82.3% [36] and 72.8% [37]. This difference in *E. coli* isolation could be due to the difference in the sample size, the types of samples and sampling area used in the research. For instance, the result from [37] includes environmental and cloacal samples and [38] used only 88 chicken cloacal samples in their study.

Antimicrobial resistance in broiler chickens has been reported in many countries in the world. This could be due to the rampant use of antibiotics as growth promoter, disease prevention and treatment of diseases in food animals. Our result revealed that most of the isolates were resistant in different degrees to commonly used antibiotics such as, tetracycline (82.5%), trimethoprim-sulphamethoxazole (78.4%) and chloramphenicol (61.9%). These results showed that most of the isolated *E. coli* from broilers in the chicken farms were multidrug-resistant. The excessive uses of these common antimicrobials in the poultry production has been reported to cause multidrug-resistance [39]. High resistance rate of tetracycline and sulfamethoxazole/trimethoprim, 91.4% and 74.2% respectively, were reported from previous study on broiler chicken farm from Malaysia [37]. In our study, some *E. coli* isolates show resistant to carbapenem antibiotics, meropenem (28.9%) and imipenem (17.5%). In this study isolates with MAR index values greater than 0.2 were 90.7%, and 9.3% were less than or equal to 0.2.

The high percentage of MAR indices with values greater than 0.2 indicates that the isolates originate from high-risk sources of contamination. This might be caused by the excessive usage of antibiotics for prevention, control of diseases, and growth promotion [40]. *E. coli* isolates were positive for ESBL encoding genes, $bla_{CTX-M}$ and $bla_{TEM}$. This finding suggests that ESBL producing *E. coli* is distributed in local broiler farms. Spread of ESBL-producing *E. coli* isolates from non-symptomatic food animals indicates that commensal *E. coli* can serve as a resistance gene reservoir and may pose a potential risk of transfer to humans [41, 42]. Previous studies from Malaysia showed high prevalence of ESBL as well as colistin resistant *E. coli* contamination from chicken meat [18–20].

In our study the dominant ESBL encoding gene was $bla_{CTX-M}$, which was also reported by several studies conducted in broilers globally [36, 38, 42–45]. Recently $bla_{CTX-M}$ is reported the leading ESBL gene worldwide. In contrast to this, a study from Germany showed $bla_{SHV}$ was the most prevalent ESBL gene in poultry [46]. We found that out of the 97 isolates of *E. coli*, $bla_{CTX-M}$ genes were detected in 62.9% of the isolates. This finding is slightly higher than previous prevalence reports from similar work by Khoshbakht R [47] which reported 60.3% prevalence of $bla_{CTX-M}$ producing *E. coli* in chicken from Iran. Similar patterns of prevalence were recently reported in Malaysia by [36] who reported that 100% (7/7) *E. coli* were positive for $bla_{CTX-M}$ genes. The presence of $bla_{TEM}$ in the *E. coli* isolated from chicken of this study was 45.4%, which is higher than a study from Iran (37.7%) [47], but it is slightly lower than *Enterobacteriaceae* isolated from surface water in Malaysia (47.4%). Similarly, it has been reported that high prevalence of $bla_{CTX-M}$ followed by $bla_{TEM}$ harboring *E. coli* recovered from human clinical samples [48, 49]. We found that 22.7% of the *E. coli* isolates were encoding both $bla_{CTX-M}$ and $bla_{TEM}$ ESBL genes, which is slightly lower than previous study from Philippines, 33.3% [42]. Evidences show that ESBL prevalence varies throughout the world where Asian countries were with the highest rates [50]. In addition, we detected the coexistence of *mcr*-1 gene with $bla_{CTX-M}$ in six of the *E coli* isolates. Which shows that, the co- resistance of another critical antibiotic, colistin. Similar findings were also reported from studies of chicken origin in Nepal and China [44, 51]. The correlation analysis of resistance antibiotic phenotypes with $bla_{CTX-M}$ and $bla_{TEM}$ genotypes did not show any correlations. Whereas, a strong correction was observed among cephalosporins, cephalosporin versus beta-lactam and beta-lactam inhibitor.

In this study the MLST sequence typing of the isolates shows that the *E. coli* isolates were widely diverse. Among the sequences, medically important groups like ST117, ST69 and ST155 were identified. ST117 and ST155 *E. coli* isolates were harboring both $bla_{CTX}$ and $bla_{TEM}$ ESBL genes. Both ST117 and ST155 were found in ESBL-producing *E. coli* from community in Malaysia [52]. ST117 was reported in extraintestinal pathogenic *E. coli* related virulence genes both in human and food-animal [53]. ST 155 were also reported in broiler chicken origin from previous research from Malaysia [20]. It was also reported with ESBL producing *E. coli* isolates in chicken meat in Singapore, Spain [54, 55] and from diseased chicken in China [56]. ST69 (ST69 complex) was also found harboring $bla_{TEM}$ gene.

Both ST117 and ST69 were reported from human EXPEC associated infections and food animals and retail meat sources in Europe [57]. In addition, ST69 was found with high virulence gene content in human EXPE in Spain [58].

## Conclusion

This study showed that prevalence of ESBL genes in *E. coli* isolates from broilers chicken farms in Malaysia is high. Our finding shows that $bla_{CTX-M}$ is the most prevalent ESBL gene in broiler farms in Kelantan, Malaysia. Most of the *E. coli* isolates were multi-drug resistant and are

therefore a potential risk as sources of ESBL producing *E. coli* infection from animals to humans by direct or indirect consumption of the animal products. The findings provide an insight that ESBL producing *E. coli* is likely spreading among the local chicken farms in Malaysia, particularly Kelantan. The spread of this multidrug resistant *E. coli* in food animals poses a risk of dissemination of the pathogen to humans through food chain.

## Supporting information

**S1 Data. Contains all supplementary data.**
(CSV)

**S1 File. Contains all the supporting figures.**
(DOCX)

**S2 File. Contain the correlation matrix.**
(IPYNB)

## Author Contributions

**Data curation:** Mulu Lemlem, Susmita Seenu Devan.

**Formal analysis:** Mulu Lemlem.

**Funding acquisition:** Erkihun Aklilu.

**Investigation:** Mulu Lemlem, Susmita Seenu Devan.

**Methodology:** Mulu Lemlem.

**Supervision:** Erkihun Aklilu, Maizan Mohammed, Fadhilah Kamaruzzaman, Zunita Zakaria, Azian Harun.

**Writing – original draft:** Mulu Lemlem.

**Writing – review & editing:** Erkihun Aklilu, Maizan Mohammed, Fadhilah Kamaruzzaman.

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
