## [Decision Letter · Decision Letter 0]

16 Jan 2023

PONE-D-22-25669Molecular Detection and Antimicrobial Resistance Profiles of Extended-Spectrum Beta-Lactamase (ESBL) Producing Escherichia coli in Broiler Chicken Farms in MalaysiaPLOS ONE

Dear Dr. Desta,

Thank you for submitting your manuscript to PLOS ONE. After careful consideration, we feel that it has merit but does not fully meet PLOS ONE’s publication criteria as it currently stands. Therefore, we invite you to submit a revised version of the manuscript that addresses the points raised during the review process.

ACADEMIC EDITOR: Point wise reply/rebuttal of the reviewer's comments should be done

We look forward to receiving your revised manuscript.

Kind regards,

Indranil Samanta

Academic Editor

PLOS ONE

and https://journals.plos.org/plosone/s/file?id=ba62/PLOSOne_formatting_sample_title_authors_affiliations.pdf.

“The authors would like to thank the Faculty of Veterinary Medicine, Universiti Malaysia Kelantan for providing research facility to conduct the research. This research was funded by Ministry of Higher Education Malaysia (MOHE) under the fundamental research grant scheme (FRGS), grant no. R/FRGS/A0600/00553A/005/2019/00700.”

“The authors would like to thank the Faculty of Veterinary Medicine, Universiti Malaysia Kelantan for providing research facility to conduct the research. This research was funded by Ministry of Higher Education Malaysia (MOHE) under the fundamental research grant scheme (FRGS), grant no. R/FRGS/A0600/00553A/005/2019/00700 awarded to Dr Erkihun Aklilu. The funders had no role in study design, data collection and analysis, decision to publish, or preparation of the manuscript.”

Additional Editor Comments:

Point wise reply/rebuttal of the reviewer's comments should be made

Reviewers' comments:

Reviewer's Responses to Questions

**Comments to the Author**

1. Is the manuscript technically sound, and do the data support the conclusions?

Reviewer #1: Yes

Reviewer #2: Yes

2. Has the statistical analysis been performed appropriately and rigorously? 

Reviewer #1: Yes

Reviewer #2: No

3. Have the authors made all data underlying the findings in their manuscript fully available?

Reviewer #1: Yes

Reviewer #2: No

4. Is the manuscript presented in an intelligible fashion and written in standard English?

Reviewer #1: Yes

Reviewer #2: Yes

5. Review Comments to the Author

Reviewer #1: The following corrections are to be made.

Line 17- excessive may be exchanged with overuse.

Line 38: Modern food animal production system requires large amounts

Line 49: also found in Salmonella

Line 53: directly or indirectly be transferred to humans through (happens vice versa)

Line 66: from chicken farm but there is no

Line 68: Therefore, this study was aimed

Line 75: in Amies transport medium and labeled

Line 80: 24 hrs.

Line 82: 24 hrs.

Line 83: confirmation by using biochemical characteristics

Line 86: Agar

Line 114: 4 min,

Line 115: 60 sec and 10 mins

Line 122: The primers used are available in (correct the language)

Line 137: isolates will be in non italic

Line 153: Our study showed that

Line 179: These results showed that most

Line 198: We found that out of the 97 isolates of E. coli,

Please check all references font and style.

Reviewer #2: The above-entitled manuscript by Mulu Lemlem Desta et al., evaluated the antimicrobial resistant profiles of Escherichia coli bacteria isolated from broiler chicken farms in Malaysia.

The manuscript is well-written. I suggest the following points to be considered before deciding this manuscript for publication:

1. Page 11, Line 102-03: Please provide brief description how Multiple antibiotic resistance (MAR) index was analyzed.

2. Some anomalies of reference citation were noticed. Reference numbers # 27 and #28 are the publications shown differently. Similar anomalies might be present. Please provide some other stronger references that attest 'Pho-gene" specific E. coli detection. Please include 16s rDNA gene sequence to confirm E. coli.

3. Please provide a supplementary figure stating cultural, biochemical characteristics, antimicrobial susceptibility analyses, and Polymerase chain reaction results for different genes described in the manuscript.

4. At page 14, line 140-41, described higher resistance of the E. coli isolates against meropenem (28.9%) than that of imipenem (17.5%) is unusual. Need to reinvestigate the findings.

5. Table 2 has described both the resistant and susceptible frequency. As resistant frequency will be the complementary of the susceptible frequency considering the total number tested, describing both frequencies is unnecessary.

6. Page 15-16, line 150-162, as part of ESBL gene analysis only two genes, bla-TEM and bla-CTXM were analyzed.

a) Some more genes should be included to draw conclusions about ESBL associated resistance.

b) There was no description about the type of bla-CTXM (type-9/ 15?).

c) There should be statistical analyses to see the association of the bla-TEM and bla-CTXM genes with the bacterial phenotypic resistance phenomena.

d) Not clear why colistin resistance gene, mcr-1 was analyzed under the titleship of ESBL. Need justification at the introduction and discussion.

7. For performing AST, authors should use at least one or two ATCC controls to validate and interpret their result.

8. This study lacks novelty as many other previous studies reported bla-TEM and bla-CTXM in poultry samples all over the globe.

6. PLOS authors have the option to publish the peer review history of their article (what does this mean?). If published, this will include your full peer review and any attached files.

Reviewer #1: **Yes: **Ripan Biswas

Reviewer #2: **Yes: **Professor. Dr. Md. Salequl Islam.

---

## [Author Response · Author response to Decision Letter 0]

20 Apr 2023

Reviewer #1:The following corrections are to be made.

Line 17- excessive may be exchanged with overuse.

Response , thank you for the suggestion corrected accordingly 

Line 38: Modern food animal production system requires large amounts

Response , thank you for the correction corrected accordingly 

Line 49: also found in Salmonella.

Response, Thank you for the suggestion and corrected accordingly

Line 53: directly or indirectly be transferred to humans through (happens vice versa)

Response , Thank you for the suggestion corrected accordingly 

Line 66: from chicken farm but there is no

Response, thank you for the suggestion corrected accordingly 

Line 68: Therefore, this study was aimed

Response, thank you for the correction and corrected accordingly 

Line 75: in Amies transport medium and labeled

Response, thank you for the correction corrected accordingly 

Line 80: 24 hrs.

Response, thank you for the correction corrected accordingly 

Line 82: 24 hrs.

Response, thank you for the correction corrected accordingly 

Line 83: confirmation by using biochemical characteristics.

Response, thank you for the Suggestion corrected accordingly. 

Line 86: Agar

Response, thank you for the correction corrected accordingly 

Line 114: 4 min,

Response, thank you for the correction corrected accordingly 

Line 115: 60 sec and 10 mins

Response, thank you for the correction corrected accordingly 

Line 122: The primers used are available in (correct the language)

Response, thank you for the correction corrected accordingly 

Line 137: isolates will be in non italic

Response, thank you for the suggestion corrected accordingly 

Line 153: Our study showed that

Response, thank you for the suggestion corrected accordingly 

Line 179: These results showed that most

Response, thank you for the Suggestion corrected accordingly 

Line 198: We found that out of the 97 isolates of E. coli,

Please check all references font and style.

Response, thank you for the correction corrected accordingly.

---

## [Editor Report · Decision Letter 1]

2 May 2023

Molecular Detection and Antimicrobial Resistance Profiles of Extended-Spectrum Beta-Lactamase (ESBL) Producing Escherichia coli in Broiler Chicken Farms in Malaysia

PONE-D-22-25669R1

Dear Dr. Desta,

We’re pleased to inform you that your manuscript has been judged scientifically suitable for publication and will be formally accepted for publication once it meets all outstanding technical requirements.

Kind regards,

Indranil Samanta

Academic Editor

PLOS ONE

Additional Editor Comments (optional): The authors have addressed the comments sufficiently
---

## [Editor Report · Acceptance letter]

12 May 2023

PONE-D-22-25669R1 

Molecular Detection and Antimicrobial Resistance Profiles of Extended-Spectrum Beta-Lactamase (ESBL) Producing *Escherichia coli* in Broiler Chicken Farms in Malaysia 

Dear Dr. Lemlem:

I'm pleased to inform you that your manuscript has been deemed suitable for publication in PLOS ONE. Congratulations! Your manuscript is now with our production department. 

Kind regards, 

on behalf of

Dr. Indranil Samanta 

Academic Editor

PLOS ONE